# Poor Nurses’ Work Environment Increases Quiet Quitting and Reduces Work Engagement: A Cross-Sectional Study in Greece

**DOI:** 10.3390/nursrep15010019

**Published:** 2025-01-13

**Authors:** Ioannis Moisoglou, Aglaia Katsiroumpa, Aggeliki Katsapi, Olympia Konstantakopoulou, Petros Galanis

**Affiliations:** 1Faculty of Nursing, University of Thessaly, 41500 Larissa, Greece; iomoysoglou@uth.gr; 2Clinical Epidemiology Laboratory, Faculty of Nursing, National and Kapodistrian University of Athens, 11527 Athens, Greece; aglaiakat@nurs.uoa.gr (A.K.); olykonstant@nurs.uoa.gr (O.K.); 3Euro-Mediterranean Institute of Quality and Safety in Healthcare, 10678 Athens, Greece; akatsapi@eiqsh.eu

**Keywords:** work environment, quiet quitting, work engagement, work, nurses

## Abstract

**Background/Objectives:** The nursing work environment, encompassing accessible resources and established processes, might affect nurses’ professional behavior. Our aim was to examine the effect of nurses’ work environments on quiet quitting and work engagement among nurses. **Methods:** We performed a cross-sectional study with nurses in Greece. We used the “Practice Environment Scale-5” to measure nurses’ work environments, the “Quiet Quitting Scale” to measure quiet quitting, and the “Utrecht Work Engagement Scale-3” to measure work engagement among nurses. We developed multivariable regression models adjusted for gender, age, understaffed wards, shift work, and work experience. **Results:** The study population included 425 nurses. The mean age of the nurses was 41.1 years. After controlling for confounders, we found that lower nurse participation in hospital affairs, less collegial nurse–physician relationships, worse nursing foundations for quality of care, and lower levels of nurse manager ability, leadership, and support were associated with higher levels of quiet quitting among nurses. Moreover, our multivariable analysis identified a positive association between nurse manager ability, leadership, and support, collegial nurse–physician relationships, nursing foundations for quality of care, and work engagement among nurses. **Conclusions:** Our findings highlight the poor work environment, elevated levels of quiet quitting, and moderate work engagement among nurses. Moreover, we found that a poor nurses’ work environment was associated with higher levels of quiet quitting. Moreover, our findings showed that nurses’ work environments had a positive impact on work engagement. The ongoing endeavor to enhance all aspects of nurses’ working conditions by healthcare organization administrations is essential for optimizing nurses’ performance, facilitating organizational operations, and ensuring service quality.

## 1. Introduction

Nurses are the largest professional cohort within a healthcare organization, and the quality and safety of the services rendered frequently hinge on their contributions [1,2]. Donabedian’s model posits that the structure and processes of a healthcare organization impact the outcomes of health service delivery [3]. Nurses are the most important resource (structure) of the healthcare system, and through the implementation of nursing interventions (process) they contribute decisively to its outcomes. However, the working environment of nurses can also influence the outcomes for nurses themselves, and indeed nurse and organizational outcomes occur concurrently [4]. The influence of the work environment on nurses’ professional conduct is not a fresh issue but has afflicted healthcare systems for decades. As early as the 1980s, the first study exploring the effect of their working environment on the recruitment and retention of nurses in hospitals was recorded, as there was a large shortage of nurses due to nursing dropouts. The survey revealed 41 hospitals, designated as Magnet Hospitals, notable for their effectiveness in retaining and recruiting nursing staff. The criteria that designated these hospitals as magnets encompassed a series of structural and process factors of nurses’ work environments, such as management and leadership style, staffing, opportunities for professional development, organizational structure (including nursing participation in hospital committees), personnel policies (such as promotion opportunities), quality of patient care (including the implementation of professional practice models), and education (emphasis on the value of education and teaching by nurses) [5].

While the literature previously focused on nurses’ turnover intention, the COVID-19 pandemic has introduced a novel concept pertaining to a new work behavior known as “quiet quitting”. The notion gained widespread recognition with a short video on the social media platform TikTok. Subsequently, the management consulting company “Gallup” carried out research in the US revealing that 50% of business sector employees have engaged in quiet quitting [6]. To attain a work–life balance, employees engage in quiet quitting by diminishing their performance and fulfilling only the minimum job criteria to evade termination, instead of exceeding expectations, arriving early, or working overtime [7]. The problem is notably prevalent in the healthcare sector, with nurses choosing quiet quitting at a rate of 67%, much above that of healthcare workers [8]. The phenomenon is very novel, and a reliable and valid measurement tool has just lately been created, resulting in restricted studies within the health sector [9]. Factors contributing to the quiet quitting among nursing personnel include burnout and bullying at work, whereas moral resilience, emotional intelligence, and job satisfaction serve as protective elements [10]. Additional reasons identified as contributing to nurses’ quiet quitting include workplace environment challenges and inadequate management practices, such as bullying, horizontal aggression, white-anting, and undermining, along with a failure to acknowledge and adequately reward the contributions of the most productive nurses [11]. Nurses who engage in quiet quitting are more prone to express turnover intentions [10]. Research on the impact of aspects of nurses’ work environments on the prevalence of quiet quitting is notably scarce [12]. In the healthcare sector, proposed interventions to mitigate quiet quitting encompass enhancing nurse autonomy and decision-making, providing professional development opportunities, fostering open communication between leadership and nurses to address their concerns, and establishing a work environment where nurses feel valued as integral team members pursuing a significant objective [7,11]. Research outside the health sector has identified several predictors of quiet quitting, including poor management, lack of specific policies such as support systems for new mothers and provision for health and mental well-being, ineffective leadership or nepotism among superiors, disparities in salary and compensation, and role conflicts that negatively impact well-being and exacerbate burnout, ultimately resulting in quiet quitting [6,13,14].

A positive work behavior is that of work engagement, which can be defined as “*a positive, fulfilling, work-related state of mind that is characterized by vigor, dedication, and absorption*” [15]. An employee exemplifying vigor, dedication, and absorption demonstrates high energy levels, mental resilience, and a commitment to exerting an effort at work, while also being deeply engaged and experiencing a sense of significance, enthusiasm, inspiration, pride, and challenge. A work-engaged employee is also characterized by complete concentration and a joyful immersion in their job, resulting in the rapid passage of time and difficulty in disengaging from tasks [15,16]. Nurses’ work engagement yields numerous advantages for both patients and the nurses themselves. In terms of patient outcomes, work engagement shown a favorable association with nurses’ perceived quality of care and patient satisfaction, while demonstrating a negative association with adverse event occurrences. The impact of work engagement on nurses encompasses a positive association with job satisfaction, career fulfillment, compassion satisfaction, work effectiveness, productivity, and overall well-being, alongside a negative association with burnout, compassion fatigue, and intentions to leave the profession [17,18,19,20]. The work environment significantly influences nurses’ work engagement, particularly the components comprising this environment, including nurse involvement in hospital affairs, the nursing foundation for quality care, nurse manager competence and leadership, the adequacy of staffing resources, and the nurse–physician relationship [21]. Management plays a significant role at both the ward and organizational levels, along with the leadership style employed [22]. Dominant leadership styles with positive associations have emerged as those of transformational leadership, as well as task-focused, authentic, ethical, resonant, and servant leadership styles [23,24,25].

To the best of our knowledge, this is the first study that has investigated the impact of nurses’ work environments on their quiet quitting. Moreover, we examined the association between nurses’ work environments and work engagement.

## 2. Materials and Methods

### 2.1. Study Design

A cross-sectional study was conducted with a sample of nurses in Greece. We collected our data in October 2024 through a web-based survey. In particular, we used Google Forms to create an online version of the study questionnaire, and then we posted it in nurses’ groups on Facebook. We selected Facebook because it is a social media platform that is used very often. Furthermore, nurses in Greece have created several Facebook groups, so it was easier to access a larger number of nurses. In this way, we obtained a convenience sample of nurses. We applied the following inclusion criteria: (a) nurses who have been working in clinical settings, (b) nurses who have been working at least one year, (c) nurses who understand the Greek language, and (d) nurses with a Facebook profile. Our study questionnaire was open continuously and available for response in October 2024.

We used G*Power version 3.1.9.2. to calculate the sample size in our study. Thus, considering a low effect size (f^2^ = 0.05) of nurse work environment on quiet quitting and work engagement, the number of independent variables (five predictors and five confounders), a confidence level of 95%, and a margin of error of 1%, the sample size was estimated at 370 nurses.

### 2.2. Measurements

Regarding demographic variables, we measured sex (females or males), age (continuous variable), work in understaffed wards (no or yes), shift work (no or yes), and work experience (continuous variable). Nurses self-assessed whether they work in understaffed wards.

We used the short form of the Practice Environment Scale of the Nursing Work Index (PES-NWI), the “Practice Environment Scale-5” (PES-5), to measure nurse work environments [26]. The PES-NWI was developed to measure the nursing practice environment, which was defined as the organizational traits that support or undermine professional nursing practice [27]. The PES-5 comprises five items that represent five dimensions of the work environment. In particular, the item “Administration that listens and responds to nurse concerns” refers to the dimension “Nurse participation in hospital affairs”, the item “A supervisor who is a good manager and leader” refers to the dimension “Nurse manager ability, leadership, and support”, the item “Good teamwork between nurses and physicians” refers to the dimension “Collegial nurse–physician relationships”, the item “Enough staff to get work done” refers to the dimension “Staffing and resource adequacy”, and the item “A clear philosophy of nursing that pervades the patient care environment” refers to the dimension “Nursing foundations for quality of care”. Answers are on a four-point Likert scale from completely disagree (1) to completely agree (4). Higher values indicate a better nurse work environment, with values greater than 2.5 indicating a supportive work environment, while values less than or equal to 2.5 indicate an unsupportive environment [27]. We used the valid Greek version of the PES-5 [28]. In our study, the Cronbach’s alpha for the PES-5 was 0.604.

We used the “Quiet Quitting Scale” (QQS) to measure levels of quiet quitting among our nurses [9]. The QQS consists of nine items, and the answers are on a five-point Likert scale from strongly disagree/never (1) to strongly agree/always (5). Some example items are the following: “I do the basic or minimum amount of work without going above and beyond”, “I find motives in my job”, and “I feel inspired when I work”. A score on the QQS is the average of the answers to the five items. Thus, a score on the QQS ranges from 1 to 5. Higher values indicate higher levels of quiet quitting. We used the suggested cut-off point of 2.06 to separate our nurses into quiet quitters and non-quiet quitters [29]. We used the valid Greek version of the QQS [8]. In our study, the Cronbach’s alpha for the QQS was 0.855.

We used the “Utrecht Work Engagement Scale-3” (UWES-3) to measure work engagement in our sample [30]. The UWES-3 comprises three items (e.g., “At my work, I feel bursting with energy”), and answers are on a seven-point Likert scale from never (0) to every day (6). A mean score on the UWES-3 ranges from 0 to 6, with higher values indicating higher levels of work engagement. We used the valid Greek version of the UWES-3 [31]. In our study, the Cronbach’s alpha for the UWES-3 was 0.812.

### 2.3. Ethical Issues

The Ethics Committee of the Faculty of Nursing, National and Kapodistrian University of Athens approved our study protocol (approval number: 01, 26 September 2024). Moreover, we took into consideration the Declaration of Helsinki when conducting our study [32]. An information sheet on the online version of the study questionnaire informed participants about the aim and the design of the study. Then, we asked participants if they consented to participate in our study. We did not provide incentives to the participants, and, thus, their participation was voluntary. Nurses with a positive answer could proceed to answering our questionnaire. We did not collect personal data from our participants, such as a name or phone number. Only study scholars had access to the Excel file that was created from Google Forms. We downloaded this file, and we secured it by adding a password.

### 2.4. Statistical Analysis

We present categorical variables as numbers and percentages. Also, we use the mean, standard deviation (SD), median, minimum value, maximum value to present continuous variables. We used the Kolmogorov–Smirnov test and Q-Q plots to examine the distribution of continuous variables. We did not recognize ceiling or floor effects on our data. The five dimensions of the PES-5 were our independent variables, while scores on the QQS and UWES-3 were our dependent variables. Since the dependent variables were continuous variables that followed a normal distribution, we applied a linear regression analysis. First, we performed a univariate linear regression analysis, and then we created a final multivariable linear regression model including all the independent variables. We adjusted the multivariable models for gender, age, understaffed wards, shift work, and work experience. We present unadjusted and adjusted beta coefficients, 95% confidence intervals (CIs), and *p*-values. *p*-values less than 0.05 were considered as statistically significant. We used IBM SPSS 21.0 (IBM Corp, released 2012, IBM SPSS Statistics for Windows, Version 21.0., Armonk, NY, USA: IBM Corp.) for statistical analysis.

## 3. Results

### 3.1. Demographics

The study population included 425 nurses. The mean age of the nurses was 41.1 years (SD; 10.0 years). In our sample, 88.9% (n = 378) were females, while 11.1% (n = 47) were males. Among our nurses, 72.0% (n = 306) reported that they work on shifts, while 82.1% (n = 349) reported that they work on understaffed wards. The mean years of clinical experience was 16.5 years (SD; 13.9 years).

### 3.2. Study Scales

Table 1 shows descriptive statistics for the study scales. The nurses’ work environment was considered unsupportive to professional nursing practice as indicated by collegial nurse–physician relationships (mean; 2.48, SD; 0.71), nurse manager ability, leadership, and support (mean; 2.29, SD; 0.86), and nursing foundations for quality of care (mean; 2.16, SD; 0.76). Also, staffing and resource adequacy (mean; 1.66, SD; 0.76) and nurse participation in hospital affairs (mean; 1.64, SD; 0.67) indicated low levels of a supportive work environment for nurses. Applying the suggested cut-off point for the QQS, 66.1% (n = 281) of our nurses could be considered as quiet quitters, while 33.9% (n = 144) could be considered as non-quiet quitters. The mean score on the QQS was 2.40 (SD; 0.73). The mean score on the UWES-3 indicated moderate levels of work engagement among our nurses (mean; 3.53, SD; 1.59).

### 3.3. Impact of Nurse Work Environment on Quiet Quitting

Table 2 presents results from the univariate and multivariable analysis, with nurse work environment as the independent variable and quiet quitting as the dependent variable. We found that a worse nurse work environment was associated with higher levels of quiet quitting. The final multivariable linear regression model showed that four out of five dimensions of nurse work environment had a negative impact on quiet quitting. In particular, lower nurse participation in hospital affairs increased levels of quiet quitting (beta = −0.116, 95% CI = −0.218 to −0.014, *p*-value = 0.026). In other words, a decrease in score in “nurse participation in hospital affairs” by one unit was associated with an increase in quiet quitting by 0.116 units. Moreover, less collegial nurse–physician relationships (beta = −0.134, 95% CI = −0.232 to −0.037, *p*-value = 0.007) and worse nursing foundations for quality of care (beta = −0.133, 95% CI = −0.229 to −0.038, *p*-value = 0.006) were associated with higher levels of quiet quitting. Thus, a reduction in score in “collegial nurse–physician relationships” and “nursing foundations for quality of care” by one unit was associated with an increase in quiet quitting by 0.134 and 0.133 units, respectively. Also, we found a negative association between nurse manager ability, leadership, and support and quiet quitting (beta = −0.177, 95% CI = −0.259 to −0.095, *p*-value < 0.001). Thus, quiet quitting was increased by 0.177 units for an increase of one unit in the score in “nurse manager ability, leadership, and support”.

Regarding demographic variables, our multivariable linear regression model showed that levels of quiet quitting were higher among males (beta = 0.240, 95% CI = 0.040 to 0.440, *p*-value = 0.019), and nurses who worked in understaffed wards (beta = 0.240, 95% CI = 0.045 to 0.435, *p*-value = 0.016). Moreover, we found a negative association between age and quiet quitting (beta = −0.009, 95% CI = −0.018 to −0.0003, *p*-value = 0.042).

### 3.4. Impact of Nurse Work Environment on Work Engagement

Table 3 shows results from the linear regression analysis, with work engagement as the dependent variable. We found that three dimensions of the nurse work environment had a positive impact on work engagement. Our multivariable linear regression analysis showed that nurse manager ability, leadership, and support (beta = 0.335, 95% CI = 0.156 to 0.514, *p*-value < 0.001), collegial nurse–physician relationships (beta = 0.391, 95% CI = 0.179 to 0.603, *p*-value < 0.001), and nursing foundations for quality of care (beta = 0.340, 95% CI = 0.131 to 0.549, *p*-value = 0.001) were associated with higher levels of work engagement among nurses. In other words, a better work environment was associated with higher levels of work engagement. In particular, an increase in score in “nurse manager ability, leadership, and support”, “collegial nurse–physician relationships”, and “nursing foundations for quality of care” was associated with an increase in work engagement by 0.335, 0.391, and 0.340 units, respectively. There were no significant associations between demographic variables and work engagement.

## 4. Discussion

This study evaluated the nursing work environment and examined the association between work environment, quiet quitting, and work engagement. The results of our study indicated that the work environment for nurses is unsupportive, since low scores were observed across every dimension of the work environment. The majority of nurses chose quiet quitting, exhibiting low levels of work engagement, while the work environment was negatively associated with quiet quitting and favorably associated with work engagement. Concerning quiet quitting, the findings indicated that nearly two-thirds of the nurses in our study (66.1%) were classified as quiet quitters, as the average score on the silent quitting scale was 2.40, over the threshold value of 2.06 [29].

In recent years, various aspects of the nursing work environment have consistently deteriorated, as numerous studies have emphasized nursing understaffing in healthcare organizations, the resignation of nurses from the profession, management’s and leadership’s failure to support nurses, and the insufficient representation of nurses on hospital committees [33,34,35,36,37]. A study conducted by The American Association of Critical-Care Nurses indicates a prolonged decline in the nurses’ work environment [38]. The current study identifies the two lowest score dimensions as staffing and resources, and nurses’ participation in hospital issues. These findings align with the studies, emphasizing these two dimensions as the least supportive in nurses’ work environments [33,39]. The persistent understaffing of nursing departments, coupled with the immense pressure on the Greek health system during the COVID−19 pandemic, significantly burdened nurses, as evidenced by studies indicating elevated levels of burnout, dissatisfaction, and turnover intention [8,10]. The prevailing conditions have facilitated the emergence of quiet quitting, potentially impacting the quality of the health services delivered, since patient care expectations remain elevated, while nurses opt to render minimal services within the framework of quiet quitting. Also, regarding the job characteristics of the participants in the present study, it was found that the percentages of nurses choosing quiet quitting were higher in understaffed wards. The high workload resulting from understaffing, as well as the possibility of working many night shifts and the potential inability to take time off or have regular leave, leads nurses to choose quiet quitting in their attempt to balance work and personal life. In addition to the widespread withdrawal of nurses from the profession, a large proportion declare their turnover intention. The turnover intention strongly predicts actual turnover [40], suggesting that understaffing could provide a substantial challenge to the effective operation of healthcare services. Nurses employed in understaffed wards, facing elevated workloads and minimal participation in hospital affairs, exhibit diminished work engagement and satisfaction [39,41]. Insufficient job satisfaction among nurses correlates with a decreased work engagement, which subsequently results in a reduced assessment of the quality of healthcare services and heightens nurses’ propensity to depart from their positions [17]. Consequently, we ascertain that healthcare systems, organizations, and nursing personnel are ensnared in a detrimental cycle, wherein an unsupportive work environment adversely impacts nurses’ work behavior and their turnover intention, thus generating job vacancies and further deteriorating the work environment.

The present study emphasized the impact of the work environment dimension of “nurse manager ability, leadership, and support” on both nurses’ work engagement and quiet quitting. Several studies have highlighted different leadership styles that positively influence nurses’ work engagement. Nurse managers exhibiting ambidextrous leadership traits empower their staff with greater responsibility, influence, and support and guide them to innovate and question traditional top-down control. As a result, ambidextrous nurse managers are likely to foster leadership among staff nurses and enhance their work engagement [42]. Also, the transformational leadership style of nurses fosters trust and confidence among their staff, cultivates a shared sense of mission and values, articulates a clear vision, promotes innovation and creativity, and empowers staff by encouraging autonomy and voice, thereby positively influencing work engagement among nurses [25]. Furthermore, nurse leaders who adopt an empowering leadership style and disseminate information to facilitate subordinate involvement in decision-making, thereby conferring power and responsibility, while promoting accountability and fostering skill development and coaching for innovative performance, establish optimal conditions for improving nurses’ work engagement [43]. There exists a research gap concerning the association between leadership and quiet quitting, as the phenomenon of quiet quitting is recent and necessitates investigations into predictive factors. The present research is the inaugural study to establish an association between nursing leadership and nurses’ inclination towards quiet quitting. A business sector study identifies poor management as a significant predictor of quiet quitting, emphasizing the necessity to enhance the competencies of those in leadership roles and to mitigate disengagement and burnout [6]. In modern healthcare systems, characterized by elevated levels of dissatisfaction, burnout, and turnover intentions among nurses, the administrative competencies of nurse supervisors are inadequate. Leadership abilities are essential to inspire, engage, and empower nurses, to enhance retention and work engagement and minimize quiet quitting work behavior. The aforementioned leadership traits are seen as crucial elements that foster a supportive workplace for nurses and mitigate the occurrence of quiet quitting among them [7,11].

Our study had several limitations. First, we collected our data through a web-based survey. Thus, our convenience sample cannot be representative of nurses in Greece. Thus, we cannot generalize our findings, since our online survey cannot be representative of all nurses in Greece. For instance, nurses without a Facebook profile could not participate in our study. Moreover, we cannot calculate the response rate, since the number of nurses that saw our study in Facebook groups but decided not to participate is unknown. Second, we used valid tools to measure the nurse work environment, quiet quitting, and work engagement, but information bias is still probable since these tools are self-reported. Thus, the self-reported basis of our scales may introduce reporting bias into our study. Third, we adjusted our multivariable models for several confounders, but several other variables may introduce confounders in the association between nurse work environment, quiet quitting, and work engagement. For instance, marital status, having children, organizational culture, individual personality traits, and the type of hospital (public or private) may have an impact on quiet quitting and work engagement, and, thus, future studies should measure demographic and job variables like those. Fourth, the cross-sectional nature of our study did not allow us to infer a causal relationship between independent and dependent variables. Fifth, the Cronbach’s alpha for the PES-5 was 0.604, and, thus, we should recognize a reliability issue regarding this tool in our study. The Cronbach’s alpha for the PES-5 was close to the limit, and, thus, further studies should be conducted even in Greece to further establish the internal reliability of the scale. The low value for the Cronbach’s alpha may be attributed to the fact that the nurses’ practice environment is complicated and difficult to measure. We should notice that the Cronbach’s alphas for the other two scales we used (QQS and UWES-3) was very good, since they were higher than 0.81. Sixth, since we collected our data through a web-based survey, the accuracy and reliability of our data depend on the honesty of our participants. Finally, information bias could be introduced in our study from the self-assessment of confounders such as working in understaffed wards.

## 5. Conclusions

The work environment of nurses can affect their behavior, hence impacting the quality of healthcare services and the efficiency of healthcare organizations. This study emphasized the poor work environment, elevated levels of quiet quitting, and moderate work engagement among nurses. Elements of the work environment were identified as predictors of both work engagement and quiet quitting. The ongoing endeavor to enhance all aspects of nurses’ working conditions by healthcare organization administrations is essential for optimizing nurses’ performance, facilitating organizational operations, and ensuring service quality.

## Figures and Tables

**Table 1 nursrep-15-00019-t001:** Descriptive statistics for the study scales.

Scales *Factors*	Mean	Standard Deviation	Median
Practice Environment Scale-5			
*Nurse participation in hospital affairs*	1.64	0.67	2.00
*Nurse manager ability, leadership, and support*	2.29	0.86	2.00
*Collegial nurse–physician relationships*	2.48	0.71	3.00
*Staffing and resource adequacy*	1.66	0.76	2.00
*Nursing foundations for quality of care*	2.16	0.76	2.00
Quiet Quitting Scale	2.40	0.73	2.33
Utrecht Work Engagement Scale-3	3.53	1.59	3.67

**Table 2 nursrep-15-00019-t002:** Linear regression analysis with score on “Quiet Quitting Scale” as dependent variable.

Independent Variables	Univariate Model			Multivariable Model ^a^		
	Unadjusted Beta Coefficient	95% CI for Beta	*p*-Value	Adjusted Beta Coefficient	95% CI for Beta	*p*-Value
Nurse participation in hospital affairs	−0.241	−0.341 to −0.141	<0.001	−0.116	−0.218 to −0.014	0.026
Nurse manager ability, leadership, and support	−0.282	−0.358 to −0.206	<0.001	−0.177	−0.259 to −0.095	<0.001
Collegial nurse–physician relationships	−0.250	−0.345 to −0.155	<0.001	−0.134	−0.232 to −0.037	0.007
Staffing and resource adequacy	−0.107	−0.197 to −0.017	0.020	0.042	−0.060 to 0.144	0.415
Nursing foundations for quality of care	−0.298	−0.385 to −0.212	<0.001	−0.133	−0.229 to −0.038	0.006
Males vs. females	0.269	−0.050 to 0.488	0.016	0.240	0.040 to 0.440	0.019
Age	−0.010	−0.017 to −0.003	0.005	−0.009	−0.018 to −0.0003	0.042
Understaffed ward	0.255	0.076 to 0.434	0.005	0.240	0.045 to 0.435	0.016
Shift work	0.066	−0.088 to 0.220	0.402	0.047	−0.106 to 0.200	0.547
Work experience	−0.005	−0.010 to 0.0003	0.068	0.001	−0.005 to 0.007	0.754

CI: confidence interval; ^a^ R^2^ for the multivariable model = 18.3%, *p*-value for ANOVA < 0.001

**Table 3 nursrep-15-00019-t003:** Linear regression analysis, with score on “Utrecht Work Engagement Scale-3” as dependent variable.

Independent Variables	Univariate Model			Multivariable Model ^a^		
	Unadjusted Beta Coefficient	95% CI for Beta	*p*-Value	Adjusted Beta Coefficient	95% CI for Beta	*p*-Value
Nurse participation in hospital affairs	0.534	0.315 to 0.753	<0.001	0.204	−0.019 to 0.426	0.073
Nurse manager ability, leadership, and support	0.603	0.436 to 0.769	<0.001	0.335	0.156 to 0.514	<0.001
Collegial nurse–physician relationships	0.657	0.452 to 0.863	<0.001	0.391	0.179 to 0.603	<0.001
Staffing and resource adequacy	0.271	0.074 to 0.468	0.007	0.089	−0.133 to 0.312	0.432
Nursing foundations for quality of care	0.698	0.510 to 0.886	<0.001	0.340	0.131 to 0.549	0.001
Males vs. females	−0.185	−0.668 to 0.298	0.452	−0.113	−0.551 to 0.324	0.610
Age	0.026	0.011 to 0.041	0.001	0.018	−0.002 to 0.037	0.078
Understaffed ward	−0.182	−0.577 to 0.214	0.367	0.047	−0.379 to 0.472	0.830
Shift work	−0.281	−0.617 to 0.056	0.102	−0.009	−0.343 to 0.326	0.959
Work experience	0.015	0.004 to 0.026	0.006	0.004	−0.010 to 0.017	0.591

CI: confidence interval; ^a^ R^2^ for the multivariable model = 18.9%, *p*-value for ANOVA < 0.001.

## Data Availability

The data presented in this study are available upon request from the corresponding author.

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
