# Peer review of "Poor Nurses’ Work Environment Increases Quiet Quitting and Reduces Work Engagement: A Cross-Sectional Study in Greece"

_nursrep, 2025, doi:10.3390/nursrep15010019_

Round 1

Reviewer 1 Report

Comments and Suggestions for Authors

This study tries to shine some light in the filed of quiet quitting amongst nurses. Happy to see this applied in the greek healthcare system.

I have some concerns regaridng teh design. Why didnt you chose more demographics? Marital status or having childersn is found to be an important factor for burnout.... Also did you provide a definition of an understaffed ward, or was it up to the nurse to "decide"? Working on a private or public hopsital or in a hospital of an island etc may play an important role in  your aim.

You had a questioanire that ahd a relatively low Cronbach's alpha. no mention of that in limitaions or anywhare else.

In table 1 the min and max do not add much

Was there a ceiling or floor effect of your data?

Author Response

Dear Reviewer, 

Reviewer 2 Report

Comments and Suggestions for Authors

This study has the potential to make important contributions to the literature on silent resignation and work engagement. However, there is a need for adjustments in terms of generalisability, methodological details and discussion of the findings. With the elimination of these deficiencies, the study will increase its value both academically and practically.

The abstract summarises the purpose of the study, the methods used and the results obtained. However, the last sentence of the abstract should further emphasise the findings of the study and clarify its contribution to the literature. Also, some details such as the sample size used are left incomplete.

The introductory section summarises the importance of the topic and the purpose of the study in a good way. The Donabedian model and related literature are appropriately discussed. However, the relevance of the concept of ‘quiet quitting’ to the health sector could be explained in more detail.

This study is a cross-sectional survey conducted among nurses working in Greece. Data collection was carried out through an online survey in October 2024. The research design was structured to examine the effects of nurses' work environment on “quiet quitting” and work engagement.

Convenience sampling method was used in the study and the survey was shared on social media platforms. This method may reduce the likelihood of the sample representing the target population of the study and may create generalisability issues.

It is not clear how the online platforms were selected and which target audience they addressed.

It is not specified whether the survey was open for a certain period of time or continuously. Information should have been provided on the length of time the survey was open, the methods used for access and the survey participation rate (e.g. how many people reached the shared survey link and how many of them responded).

It was not clearly explained how voluntary participation was ensured and whether there was an incentive element. No information was provided on how the anonymity of the participants was ensured and the security measures of the online platform in this regard. The reason for choosing Google Forms and the advantages/disadvantages of this platform were not mentioned. The extent to which the online survey is representative of all nurses in Greece is not stated.

Data were collected through an online questionnaire. This may only include nurses with digital access and may bias the results. The accuracy and reliability of online surveys depend on the honesty of the respondents. Limitations.

The reliability coefficients (Cronbach's alpha) of the scales used were low or close to the limit (e.g. α=0.604 for PES-5). This may raise doubts about the internal consistency of the scales. The reasons for the low reliability coefficients should have been discussed and additional analyses (e.g. item analyses) should have been conducted if necessary.

The use of a cross-sectional design is not suitable for testing cause-effect relationships. This limits the interpretability of the main findings of the study.

The findings are supported by tables and expressed descriptively in the text.

Statistically significant relationships between silent resignation and work engagement and work environment are presented. What does the silent resignation score of 2.40 (SD=0.73) mean for practice?

Among the findings, the effect of demographic data on dependent variables is not sufficiently included. For example, the effects of age, gender or work experience on silent resignation or work engagement are not analysed in detail.

The regression results presented in Table 2 and Table 3 are not sufficiently explained. For example, more information should be provided on the magnitude and direction of the effects of the beta coefficients.

The statistical significance of the beta coefficients (e.g. -0.116, -0.134) is indicated, but the practical implications of these effects are not discussed.

The discussion section compares the findings of the study with the literature and discusses the impact of the work environment on nurse behaviour in general.

The discussion made links with the existing literature and evaluated the consistency of the findings with previous studies. It was underlined that the concept of ‘silent resignation’ is a new phenomenon in the health field and the originality of this study was emphasised.

The positive impact of leadership styles on work engagement and the role of management in improving the working environment were emphasised. Limitations of the study were clearly stated (e.g. cross-sectional design, convenience sampling method).

It is not discussed why the silent resignation rates (66.1%) are so high. How these rates can be attributed to the health care system, cultural factors and working conditions has not been emphasised. The relationship between silent resignation rates and the workload of nurses in Greece, deficiencies in the health system and cultural factors should be elaborated

The contribution of the study to the literature is not emphasised strongly enough. For example, the relationship between silent resignation and leadership styles was presented as a new finding, but the details of this relationship were not discussed in depth.

The discussion superficially addressed the impact of limitations and did not explain the potential impact of these limitations on the findings.

The data were collected online through Facebook groups, which limits the sample of the study to be unrepresentative of the general population of nurses.

All of the scales used were self-report based, which may lead to reporting bias.

The cross-sectional design of the study does not make it possible to establish a cause-and-effect relationship.

The study controlled for some confounding variables (e.g. age, gender, shift system). However, other potential influencing factors (e.g. organisational culture, individual personality traits) were not taken into account.

The Cronbach's alpha value of the PES-5 scale used showed a value close to the limit with 0.604.

This indicates that the internal consistency of the scale is low and may reduce the reliability of the results.

Although the validity of the Greek versions of the scales was reported, no details about the cultural adaptation process were provided.

Author Response

Dear Reviewer, 

Round 2

Reviewer 1 Report

Comments and Suggestions for Authors

Thanks for the revisions!

Reviewer 2 Report

Comments and Suggestions for Authors

As a result of the revisions, the text has been improved to significantly enhance the scientific contribution and applicability of the study. In the discussion section, the findings of the study are discussed with a stronger connection to the existing literature and the effects on silent resignation and work engagement are comprehensively evaluated. Gaps in the literature are highlighted and the unique contribution of this study in this field is clearly stated. In particular, the effects of leadership styles and management skills on reducing silent resignation and increasing work engagement are discussed in detail.

In the revisions, solution suggestions were added and these suggestions provided concrete ways to prevent silent resignation, increase employee satisfaction and strengthen work engagement. Furthermore, the limitations of the study are clearly stated. For example, issues such as the impact of the online data collection method on representativeness and the lack of measurement of some demographic factors were addressed. In addition, recommendations for future research are provided, indicating potential directions to further enhance the contribution of the study to the field.

The text clearly demonstrates how the results can be transferred to health sector practices and emphasises that the development of leadership skills in particular can have positive effects at both individual and organisational levels. In this context, factors such as the effects of silent resignation, workload, staff shortage are discussed in detail and solutions are presented in this context. With the revisions, the text has become stronger and more comprehensive in both academic and practical terms. In this form, the study is at an acceptable level and suitable for publication.